# Thiocarbohydrazones Based on Adamantane and Ferrocene as Efficient Corrosion Inhibitors for Hydrochloric Acid Pickling of C-Steel

**Abdelwahed R. Sayed [1,2] and Hany M. Abd El-Lateef [1,3,***

[1] Department of Chemistry, College of Science, King Faisal University, P.O. Box 400, Al-Ahsa 31982, Saudi Arabia; arsayed@kfu.edu.sa
[2] Chemistry Department, Faculty of Science, Beni-Suef University, Beni-Suef 62514, Egypt
[3] Chemistry Department, Faculty of Science, Sohag University, Sohag 82524, Egypt
[*] Correspondence: hmahmed@kfu.edu.sa or hany_shubra@science.sohag.edu.eg

**Abstract:** N'-(adamantan-2-ylidene)hydrazinecarbothiohydrazide and 2-(ferrocenyl-1-ylidene) hydrazinecarbothiohydrazide are used in coordination and organometallic complexes. The important idea of the research in this paper is the principal to prepare thiocarbohydrazones from the reaction of 2-acetylferrocene (Fe-Th) or 2-adamantanone (Ad-Th) with carbonothioic dihydrazide. The materials were elucidated by elemental analysis and spectral data. The as-prepared compounds were applied as effective corrosion inhibitors for HCl pickling of C-steel. Detailed investigations on electrochemical (open circuit potential (OCP) vs. time, potentiodynamic polarization (PDP), and impedance spectroscopy (EIS)) techniques and surface morphology studies are introduced in this work. Results indicated that Fe-Th could deliver greater inhibition performance than Ad-Th, and the highest protection capacity values of 93.6% (Ad-Th) and 97.9% (Fe-Th) were accomplished at 200 ppm. The adsorption of Ad-Th or Fe-Th additives followed the Langmuir isotherm with both the chemical and the physical adsorption with chemisorption predominance. EIS measurements supported a betterment in the capacitive behavior with the corrosion inhibitors. The inhibitors exhibited a mixed-type behavior as observed from the PDP studies. Field emission scanning electron microscopy (FESEM) and Fourier-transform infrared spectroscopy (FTIR) studies emphasize the occurrence of a protective layer of the as-synthesized organic inhibitors on the C-steel interface. Theoretical studies (density functional theory (DFT) calculations and Monte Carlo (MC) simulations) provide appropriate support for the experimental findings. The existing report provides very significant consequences in formulating and designing novel thiocarbohydrazone inhibitors with high protection efficacy.

**Keywords:** ferrocene; corrosion inhibition; surface morphology; EIS; thiocarbohydrazones; DFT calculations

## 1. Introduction

Currently, carbon steel (C-steel) is applied in various manufacturing processes, including cooling water systems, oil and gas processing industry, refining, water pipes, skyscrapers, and boilers. Its wide applications are due to accessibility, low-cost, and simplicity of production. Nonetheless, C-steel remains susceptible to various corrosion forms [1–3]. In numerous industrial procedures—like well acidizing, acid pickling, and acid cleaning—contaminated and rust scales were usually extracted using acid mediums. Furthermore, HCl can be produced as a by-product in the crude oil desalting process and some oil refinery treatments [4]. HCl is the most extensively applied in the metals pickling processes [5]. Actually, corrosion is a dangerous process that decreases material features and makes them impracticable.

In this regard, the application of organic additives is a practical alternative to protect alloys and metals from corrosion in an aggressive medium [4]. The capacity of inhibitor molecules depends mainly on the metal interface, the nature of the solution, and the inhibitor structure [6]. Indeed, the research in the branch of corrosion protection by inhibitors is focused on the preparation of ecofriendly inhibitors. The inhibitors having hetero-atoms (sulfur, nitrogen, phosphorus, oxygen, etc.), functional groups, π-electrons, and benzene rings which not only have the capability to adsorb on a huge surface area but could interact with the steel interface through physical or chemical adsorption [6]. The great molecular weight compounds containing a high part of surface coverage are proven to be outstanding candidates for corrosion inhibitors [7].

Recently, thiocarbohydrazones were obtained from the reaction of salicylaldehyde derivatives with thiocarbohydrazide [8]. Synthesis of thiocarbonohydrazone containing tin for studying biological and crystal structures was reported [9]. Preparation and quantum calculations for bis-(isatincarbohydrazones or isatinthiocarbohydrazones) are described [10]. Mono- and bis-(thiocarbohydrazones) based on 2-acetylpyridine have antioxidant activity are described [11]. Preparation of thiocarbohydrazones containing benzofuran heterocycles was reported [12]. Thiocarbohydrazone-1-(4-arylidene)-5-(2-oxoindolin-3-ylidene) having antidiabetic properties was explained [13]. Schiff bases of thiocarbohydrazide are studied as chemical physio complexes [14]. Syntheses of isatin-β-thiocarbohydrazones based on the microwave-assisted are evaluated against antiquorum-sensing, antimicrobial, and cytotoxic activities [15]. Thiocarbohydrazones based on 2-nitro benzaldehyde helped in the extraction for the determination of ruthenium (III) in catalysts and alloy [16]. Thio-ligands are used in spectrophotometric in the detection of tellurium (IV) and liquid–liquid extraction [17].

The reaction of malononitrile dimer with thiocarbohydrazones gave thiadiazine derivatives based on the microwave-assisted as green chemistry [18]. Syntheses of coumarin by using simple and efficient methods were prepared as fluorescent chemosensor for fluoride detection [19]. The reaction of bisthiocarbohydrazones with cis-dioxomolybdenum (VI) gave complex for electrochemical applications [20]. Enhanced corrosion inhibition of carbon steel in HCl solution by synthesized hydrazine derivatives was reported [21–24].

The aim of this work is the examination of the corrosion protection performance of some thiocarbohydrazones based on adamantane and ferrocene in the hydrochloric acid medium for C-steel. Thiocarbohydrazones compounds are characterized through FTIR and nuclear magnetic resonance (NMR) spectroscopies. Furthermore, the influence of the well-designed groups on the protection action is experimentally inspected by the OCP vs. time, PDP, and EIS techniques as well as the surface morphology (FESEM, FTIR). Moreover, the thermodynamic indices of the adsorption isotherms were investigated to indicate the adsorption type of Ad-Th and Fe-Th on the metal surface. Furthermore, the compositional support of the findings was carried out with density functional theory (DFT) and Monte Carlo (MC) simulations to authenticate the experimentally obtained outcomes.

## 2. Experimental Part

### 2.1. Instrument, Solutions, and Materials

All required solvents and chemicals from Aldrich were analytical grade and were used without additional purification. The melting points were determined on an Electro-thermal IA 9000 sequence digital Device of melting point (Bibby Sci. Lim. Stone, Staffordshire, UK). FT-IR spectra were recorded by using FT-IR 8101 PC spectrophotometers (Shimadzu, Tokyo, Japan). $^1$H spectra were measured at 300 MHz in deuterated dimethyl sulfoxide (DMSO-d6) (VX-300 NMR spectrometer; Varian, Inc., Karlsruhe, Germany). Mass spectra were measured on a GCMS-QP1000 EX mass spectrometer (Shimadzu, Tokyo, Japan) at 70 eV. A German-made Elementarvario LIII CHNS analyzer (CKIC, Frankfurt, Germany) determined elemental analyses.

The composition (wt%) of the designated C-steel sample applied as a working electrode in the current study was carbon (0.19%), silicon (0.18%), manganese (0.05%), chromium (0.75%), nickel (0.18%), and iron balance. The protective action of the investigated inhibitors was considered at five different concentrations quantified by 20, 40, 80, 150, and 200 mg L$^{-1}$.

## 2.2. Synthesis of N′-(Adamantan-2-Ylidene)Hydrazinecarbothiohydrazide (4) and 2-(Ferrocenyl-1-Ylidene) Hydrazinecarbothiohydrazide (5)

Equimolar amounts of the appropriate carbonothioic dihydrazide 1 with 2-adamantanone 2 or 2-acetylferrocene 3 in ethanol (40 mL) were heated for 4.5 h. The resulting solids were cooled and crystallized from MeOH to provide 4 and 5, as shown in Scheme 1.

**Scheme 1.** Synthesis procedure of thiocarbohydrazones Ad-Th and Fe-Th.

N′-(Adamantan-2-ylidene)hydrazinecarbothiohydrazide (4): White solid; Yield (88%); mp > 217 °C. IR (KBr): $\nu_{max}$ 3266, 3177 (NH, NH$_2$), 2986 (aliphatic), 1108 (C=S) cm$^{-1}$. $^1$H NMR (DMSO-$d_6$): 1.08–1.88 (m, 6H, adam), 2.10 (m, 8H, adam), 4.47 (s, 2H, NH$_2$), 7.72 (s, 1H, NH) and 8.96 (s, 1H, NH) ppm. MS $m/z$ (%): 238 (M$^+$, 21). Anal. Calcd for C$_{11}$H$_{18}$N$_4$S (238.35): C, 55.43; H, 7.61; N, 23.51. Found: C, 55.41; H, 7.63; N, 23.52%.

2-(Ferrocenyl-1-ylidene) hydrazinecarbothiohydrazide (5): Pale yellow solid; Yield (80%); mp: 188 °C. IR (KBr): $\nu_{max}$ 3265, 3177, (NH, NH$_2$), 2909, 2884, 2849 (aliphatic), 1091 (C=S) cm$^{-1}$. $^1$H NMR (DMSO-$d_6$): 2.19 (s, 3H, CH$_3$), 4.18–4.24 (m, 5H, C$_5$H$_4$), 4.36–4.64 (m, 4H, C$_5$H$_4$), 4.83 (s, 2H, NH$_2$), 9.02 (s, 1H, NH) and 9.83 (s, 1H, NH) ppm. MS $m/z$ (%): 316 (M$^+$, 19); Anal. Calcd for C$_{13}$H$_{16}$FeN$_4$S (316.04): C, 49.38; H, 5.10; N, 17.72. Found: C, 49.36; H, 5.11; N, 17.70%

## 2.3. Experimental Setup and Corrosion Measurements

All the experimentations were accomplished in a 1-L glass cell at atmospheric pressure. The cell comprised of a distinctive three-electrode shape where a C-steel was applied as a working electrode, Ag/AgCl/KClsat electrode was utilized as a reference and a Pt sheet electrode served as a counter. Corrosion performance of the C-steel in molar hydrochloric acid solution with the occurrence of organic additives was analyzed by Gamry Potentiostat/Galvanostat/ZRA (Gamry Instruments, Warminster, PA, USA). The electrochemical determining was documented after reaching equilibrium conditions at open circuit potential ($E_{OCP}$). EIS measuring was used a frequency break of 10 mHz to 100 kHz at $E_{OCP}$ over. In addition, PDP evaluation was measured by altering the potential of the C-steel from −250 to +250 mV based on $E_{OCP}$ at a sweep rate of 0.2 mV/s [25]. Before accomplishing the experiment, to obtain a steady-state open circuit potential ($E_{OCP}$), the C-steel was exposed to molar HCl medium at OCP for 50 min. Gamry applications include software DC105 (version 4.35) for corrosion, EIS300 (version 5.50) for EIS measurements, and Echem Analyst 6.0 software package

(Gamry Instruments, Warminster, PA, USA) for data fitting. It is well documented that the corrosion rate of C-steel depends on the temperature, and the maximum corrosion rate is noticed at 50 °C, the optimized corrosion temperature [26]. Consequently, to evaluate the protection capacity of the as-prepared Fe-Th and Ad-Th additives on C-steel, all the measurements were accomplished at 50 °C. Each measurement was duplicated at least three times and the values of corrosion parameters were recorded as mean ± standard deviation.

## 2.4. Surface Characterization

The surface analysis by FE-SEM (JSM 5410 Model JEOL, Tokyo, Japan) of the C-steel electrode was achieved after immersion of these samples for 48 h in 1.0 M HCl without and with 200 ppm of Fe-Th. Experimental procedures of the FTIR of adsorbed films were similar, as designated in our previous report [27].

## 2.5. Theoretical Studies (DFT Calculations and MC Simulation)

DFT calculations and MC simulation had been conducted using DMol3 and adsorption locator modules in Materials Studio software V.7.0 from Accelrys Inc. (San Diego, CA, USA). For DFT calculations, the investigated Ad-Th and Fe-Th molecules have been fully optimized using B3LYP functional (Becke three-parameters Lee–Yang–Parr) with the double numeric plus polarization (DNP) basis set and permitting the treating of solvation impacts utilizing COnductor-like Screening MOdel (COSMO) controls, all these inputs are well-defined as in Eid et al. [28]. For MC simulation, the adsorption locator reveals the potential adsorption configurations of the Ad-Th and Fe-Th molecules with Monte Carlo searches on the Fe (110) surface to assess the inhibition capacity of additives [29]. The interactions of Ad-Th and Fe-Th and Fe surface (110) accomplished in a simulation box (32.27 Å × 32.27 Å × 50.18 Å) with periodic boundary conditions [29]. Forcite classical simulation engine was utilized to optimize the energy of Ad-Th and Fe-Th molecules. To the establishment, the Ad-Th and Fe-Th molecules/C-steel corrosion system in aqueous media the layer builder was implemented, and this system includes the optimized Fe (110) surface, water, and the inhibitor molecules. The universal simulation studies with force field were operated to simulate the adsorption performance of Ad-Th and Fe-Th molecules on the surface of iron (110) [30].

## 3. Results and Discussions

### 3.1. Structure Configuration of the As-Prepared Compounds

Treatment carbonothioic dihydrazide 1 with 2-adamantanone 2 or acetylferrocene 3 in ethanol under heating produced the final products N′-(adamantan-2-ylidene)hydrazinecarbothiohydrazide 4 and 2-(ferrocenyl-1-ylidene) hydrazinecarbothiohydrazide 5, respectively. The compounds 4 or 5 have been synthesized in suitable yield (Scheme 1). The compounds 4 or 5 were elucidated by spectral data and elemental analysis. The obtained data are compatible with the proposed structures for these compounds as depicted in the experiment part, for instance, the IR spectrum 4 or 5 exhibited no absorption bands at 1850–1650 cm$^{-1}$ (Figure 1) which is belong to C=O group and this elucidated the loss of water molecule to give thiocarbohydrazones 4 or 5.

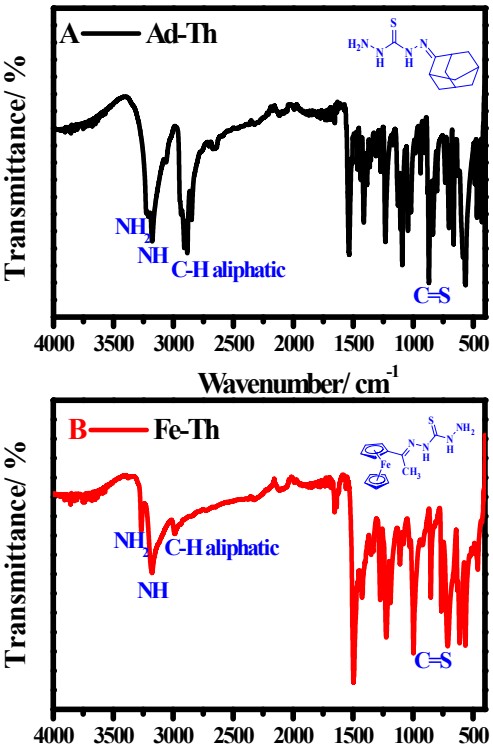

**Figure 1.** FTIR spectra of the as-prepared (**A**) Ad-Th and (**B**) Fe-Th compounds.

### 3.2. OCP vs. Time and PDP Studies

The required time to extend the values of steady OCP was obtained from the test measuring the values of OCP vs. immersion time for a C-steel electrode in the existence or lack of 200 ppm of Ad-Th and Fe-Th inhibitors in molar HCl solution. As displayed in Figure 2, the $E_{OCP}$ values obtained in blank medium, changed partially towards the positive direction, extended steady-state in the closest time, resulting in the free $E_{cor}$ of the C-steel. However, in the presence of 200 ppm Ad-Th and Fe-Th inhibitors, the $E_{OCP}$ values shifted slowly towards the negative direction and stretched a stable state after ~20 min. In the case of the Fe-Th compound, the OCP value shifted towards a more negative value, this phenomena is related to inhibitor adsorption or/and deposition of the products of the corrosion reaction on the C-steel [25]. The magnitude of the OCP shift indicates that Ad-Th and Fe-Th inhibitors at the same time affect the cathodic and anodic reactions.

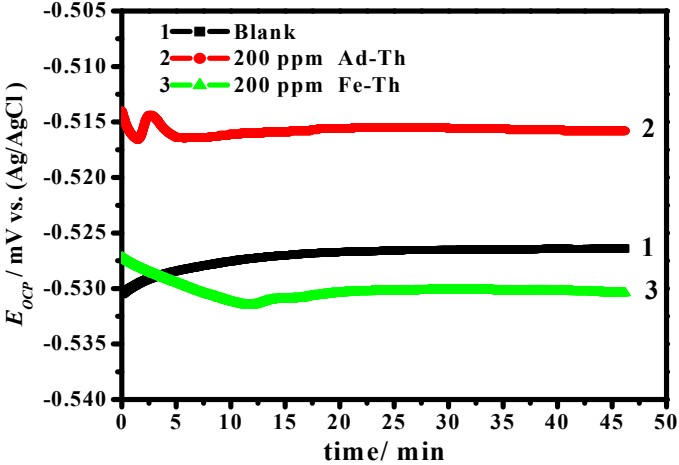

**Figure 2.** Variation of the OCP of C-steel in molar HCl with immersion time in the lack and occurrence of 200 ppm of Ad-Th and Fe-Th inhibitors at 50 °C.

The information on the electrochemical kinetics of the inhibition performance and the adsorption of corrosion inhibitors could be achieved via PDP experiment measurements [31,32]. In the current investigation, the PDP measurements were carried out in the blank and with the varying inhibitor concentrations and the outcomes are presented as the Tafel profiles in Figure 3. By inspections of Figure 3A,B, it is observed that the anodic and the cathodic current densities shifted to lower values in the presence of Ad-Th and Fe-Th inhibitors. This suggests that the addition of the Ad-Th and Fe-Th inhibitors to the aggressive solution produces a barrier film, which lowers both the cathodic hydrogen evolution and the anodic C-steel dissolution reaction. From the extrapolation of the linear segments of the Tafel diagrams, the electrochemical parameters—i.e., corrosion potential ($E_{cor}$), cathodic and anodic Tafel slopes ($\beta_c$ and $\beta_a$), and corrosion current density ($i_{cor}$)—were computed and are documented in Table 1. The corrosion inhibition capacity ($\eta_P$) and the part of surface coverage ($\theta$) were calculated using $i_{cor}$ as shown below [33]

$$\eta_P/\% = \left[ \frac{i_{cor}^B - i_{cor}^I}{i_{cor}^B} \right] \times 100\% = \theta \times 100\% \tag{1}$$

where $i_{cor}^B$ and $i_{cor}^I$ are the $i_{cor}$ in the blank and with the Ad-Th or Fe-Th inhibitors. It has been well-known in the literature [34] that if the $E_{cor}$ value of displacement in the presence of inhibitor is ≥85 mV with respect to $E_{cor}$ in blank solution, the inhibitor is characterized as anodic or cathodic-type inhibitor according to the direction of the potential shift either to the more positive or to the more negative direction, respectively [35]. If the shift is <85 mV then the inhibitor is categorized as a mixed type of nature. In the present study, the maximum change of $E_{cor}$ is less than 0.038 and 0.026 V in the presence of Ad-Th and Fe-Th inhibitors, respectively. This suggests that the inhibitors display a mixed nature [36]. The C-steel in blank HCl solutions (free inhibitor) presents an $i_{cor}$ of 387.4 µA·cm$^{-2}$. The occurrence of additives declines remarkably $i_{cor}$ to attain 24.8 and 8.1 µA·cm$^{-2}$ with 200 ppm of Ad-Th and Fe-Th, respectively. The maximum protection power for corrosion comes out to be 93.6% and 97.9% in the presence of 200 ppm Ad-Th and Fe-Th, respectively. These findings approve that the existence of either $\pi$-electrons or –NH, –C=S, –NH$_2$ groups on the inhibitor structure is promising to increase protection efficiency in the HCl solution. It is obvious from Table 1 that the $\beta_c$ values in the inhibited and uninhibited medium remain almost unchanged which indicated that the addition of as-prepared additives does not affect the hydrogen evolution mechanism, and the reduction of H$^+$ ions predominantly occurs via a charge transfer mechanism [37]. Furthermore, it could be observed from the anodic side of Figure 3A,B that the C-steel oxidation shrinkages regularly with the existence of Ad-Th and Fe-Th, at the applied anodic potentials.

The influence of additive dose from 20 to 200 mg L$^{-1}$ on the protection capacity has been examined as recorded in Table 1. Generally, it was remarked that all the organic additives provided high protection power even at a low dose. The as-prepared inhibitors, Ad-Th and Fe-Th, approve again their high protection capacity which reaches ~93.6% and 97.9%, respectively, at 200 mg L$^{-1}$. The performance of these additives could be related to the surface coverage and the stability of the protective layer formed on the steel interface even in small quantities. The additive species were deposited on the C-steel/HCl solution surface to form a protective layer by sharing the electrons between the $\pi$-electrons or –NH, –C=S, –NH$_2$ groups and the empty d-orbital of iron atoms. The higher protection capacity is attributed to the additive structure and the occurrence of various substituted groups.

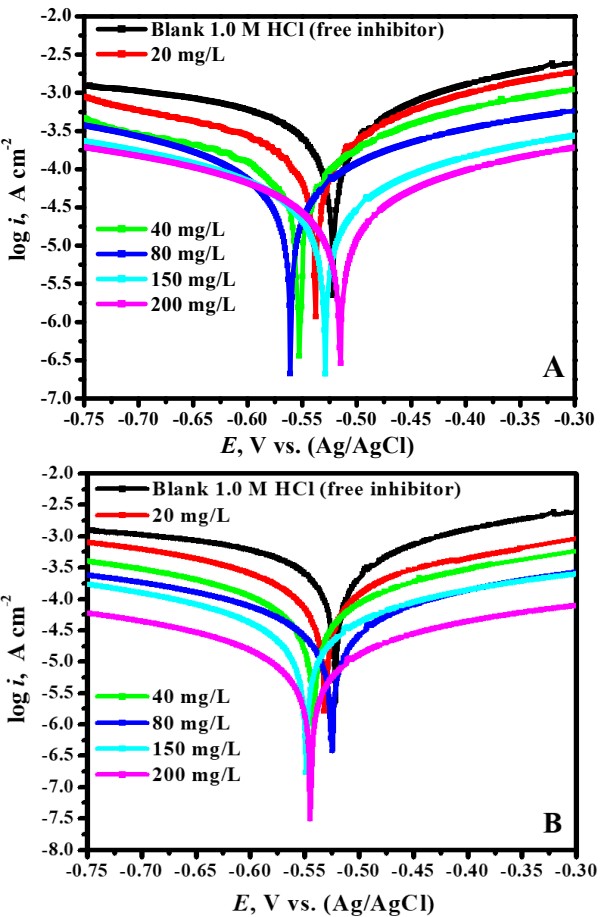

**Figure 3.** Polarization profiles for C-steel in molar HCl containing various doses of (**A**) Ad-Th and (**B**) Fe-Th at 50 °C.

**Table 1.** PDP parameters and the inhibition capacitates derived for C-steel in molar HCl medium as function of inhibitor dose at 50 °C

| Inhibitor Code | $C_{inh}/$ mg L$^{-1}$ | $i_{cor} \pm$ SD/ µA·cm$^{-2}$ | $-E_{cor} \pm$ SD/ V (Ag/AgCl) | $\beta_a/$ mV dec$^{-1}$ | $-\beta_c/$ mV dec$^{-1}$ | $\theta$ | $\eta_P/\%$ |
|---|---|---|---|---|---|---|---|
| Blank | 0.0 | 387.4 ± 26 | −0.523 ± 0.035 | 91 | 145 | – | – |
| Ad-Th | 20 | 267.7 ± 17 | −0.537 ± 0.051 | 94 | 145 | 0.309 | 30.9 |
| | 40 | 211.5 ± 14 | −0.552 ± 0.043 | 96 | 154 | 0.454 | 45.4 |
| | 80 | 156.9 ± 9 | −0.561 ± 0.053 | 92 | 157 | 0.595 | 59.5 |
| | 150 | 96.4 ± 7 | −0.529 ± 0.047 | 95 | 139 | 0.751 | 75.1 |
| | 200 | 24.8 ± 3 | −0.514 ± 0.049 | 93 | 144 | 0.936 | 93.6 |
| Fe-Th | 20 | 237.1 ± 21 | −0.530 ± 0.056 | 96 | 157 | 0.388 | 38.8 |
| | 40 | 180.1 ± 17 | −0.542 ± 0.040 | 98 | 153 | 0.535 | 53.5 |
| | 80 | 117.8 ± 10 | −0.524 ± 0.039 | 94 | 147 | 0.696 | 69.6 |
| | 150 | 53.1 ± 5 | −0.549 ± 0.051 | 98 | 148 | 0.863 | 86.3 |
| | 200 | 8.1 ± 2 | −0.544 ± 0.046 | 100 | 144 | 0.979 | 97.9 |

*3.3. EIS Studies*

The EIS experiments provided an appropriate and non-destructive means to examine the interfaces of solid metallic substrates exposed to an electrolytic solution [38]. The corrosion inhibition performance of the Ad-Th and Fe-Th molecules was investigated using the impedance investigations and the achieved findings are graphically presented in the form of the Nyquist, Bode module, and bode phase profiles in Figure 4A–C. Both in the uninhibited (free inhibitor) as well as in inhibited systems,

only one depressed semicircle was detected having their center below the real x-axis (Figure 4(I-A,I-B)). This kind of performance is characteristic of the metallic surfaces suffering the corrosive attack of the acidic medium [39]. The semicircle diameters in the Nyquist diagrams are showed to rise with increasing the inhibitor concentration, which designates the formation of a protective layer between the aggressive solution and the pristine metal [40]. Furthermore, the shape of these profiles for the inhibited systems is the same as the blank ones, indicating that the occurrence of Ad-Th or Fe-Th additives improve the impedance but does not modify the other electrochemical features [39,40]. The acquired Nyquist diagrams are not perfect semicircles but are depressed loops (Figure 4(I-A,I-B)). These deviation types are frequently encountered due to the inhomogeneity and/or surface roughness of the C-steel substrates [41].

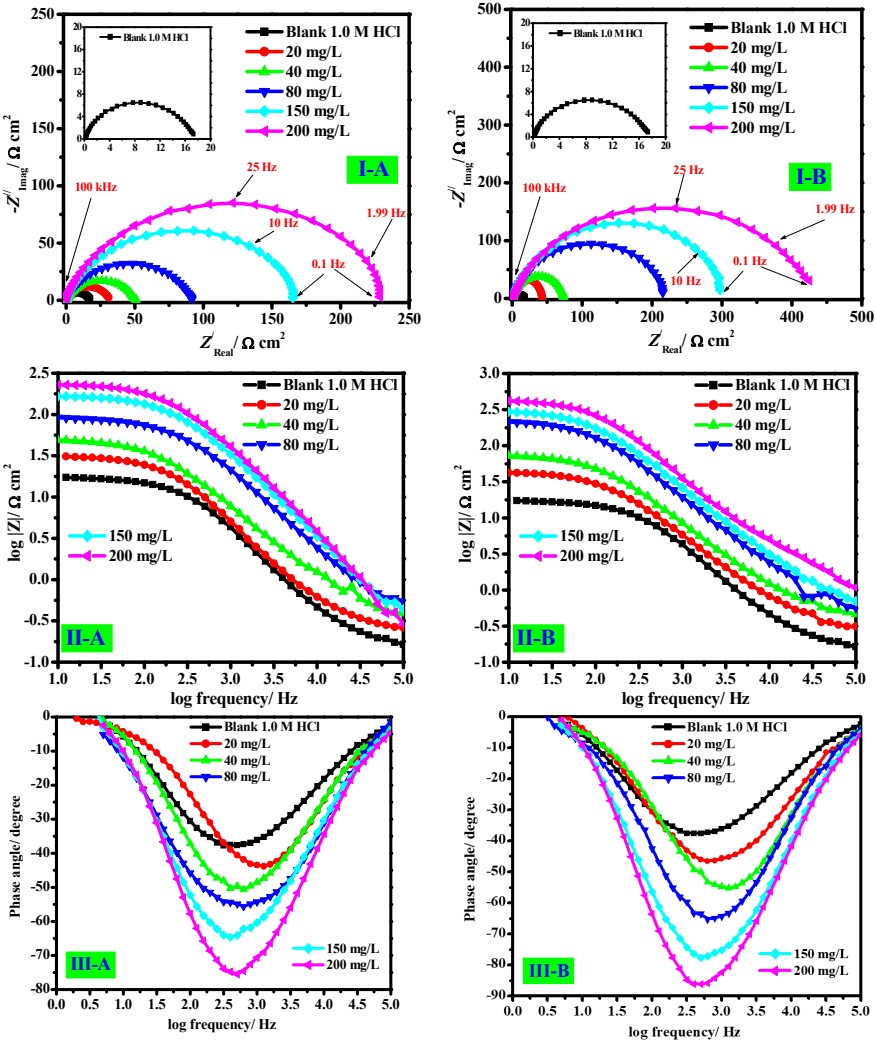

**Figure 4.** Nyquist profiles (**I**) for C-steel in molar HCl in the lack and in the occurrence of the different doses (**A**) Ad-Th and (**B**) Fe-Th at 50 °C. (**II**) Bode module and (**III**) bode phase diagrams.

The Bode and the phase angle modules for the blank and the inhibited systems are depicted in Figure 4(II-A,II-B,III-A,III-B). The diagrams obtained for all specimens (uninhibited and inhibited) confirm the presence of one time-constant route. The aberration from the perfect capacitive behavior (phase angle −90° and slope −1) could be related to the inhomogeneity of interface resulting owing to the corrosion development of the surface of C-steel [42]. Nevertheless, in the presence of as-prepared Ad-Th and Fe-Th molecules, the approach of the phase angle and the slope towards the ideal behavior could be ascribed to the corrosion protection performance of Ad-Th and Fe-Th inhibitors [43].

The EIS findings were fitted to the equivalent circuit diagram to obtain the electrochemical parameters, which are recorded in Table 2 with the protection capacity. The fitted Nyquist diagrams in the lack and with the Ad-Th and Fe-Th inhibitors are presented in Figure 5A,B. A worthy fit with the equivalent circuit diagram was achieved for all experimental findings. The equivalent circuit model using Echem Analyst software supported by Gamry electrochemical is displayed in Figure 5 (inset), in which the resistance of electrolyte ($R_s$) is shorted by a constant phase element (CPE) that is in parallel to the polarization resistance ($R_p$). The reason for using $R_p$ rather than the resistance of charge transfer ($R_{ct}$) is formerly discussed in detail [44]. In the case of the inhibited system, the $R_p$ is the collective contribution of the resistance from the $R_{ct} + R_f$ (the film resistance formed on the C-steel surface) [45]. The CPE is frequently utilized to substitute the capacitance of double-layer ($C_{dl}$) to provide a more precise fit of the experimental EIS findings. The impedance of CPE ($Z_{CPE}$) is given by [45]

$$Z_{CPE} = Y_0^{-1}(j\omega)^{-n} \tag{2}$$

where $Y_0$ symbolizes a proportionality coefficient, $j$ is the imaginary digit, $\omega$ represents the angular frequency, $n$ characterizes CPE exponent with values between 0 and 1, and delivers a measure for the surface inhomogeneity. The increase in the $n$ value (0.861–0.898) in the case of inhibited system, in comparison with the $n$ value obtained in the blank aggressive medium (0.724), could be attributed to a lessening of surface heterogeneities produced by Ad-Th and Fe-Th molecules adsorption at C-steel/medium interface. The protection capacity ($\eta_E$) was calculated as [46]

$$\eta_E/\% = \left[\frac{R_p^i - R_p^0}{R_p^i}\right] \times 100\% = \theta \times 100\% \tag{3}$$

where $R_p^0$ and $R_p^i$ denote the polarizations resistances without and containing inhibitor, respectively.

It is observed from Table 2 that the $R_p$ values can be ranked in the following sequence: Fe-Th > Ad-Th, while the $C_{dl}$ values follow the opposite order, leading to the protection capacity order as follows: Fe-Th > Ad-Th. The highest protection capacity values are 91.8% and 95.9% for Ad-Th and Fe-Th, respectively. Moreover, a significant decrease in the $C_{dl}$ values against inhibitor concentration is demonstrated. The values of $C_{dl}$ decrease from 336.1 μF cm$^{-2}$ for the C-steel in uninhibited medium to 54.0 and 39.6 μF cm$^{-2}$ in the case of 200 ppm Ad-Th and Fe-Th, respectively. The organic additive species adsorb at the C-steel/HCl interface by substituting the pre-adsorbed $H_2O$ with inhibitor molecules leading to the development of a defensive film on the metal interface. This causes a lowering of the local dielectric constant and also a thickening of the double layer. These two aspects cause an enhancement in the capacitive performance and the part of surface coverage ($\theta$) by effective adsorption of inhibitor molecules, which led to improving the protection capacity.

**Table 2.** EIS parameters and protection capacity of C-steel in 1.0 M HCl in the lack and existence of various concentrations of thiocarbohydrazones derivatives at 50 °C

| Inhibitor Code | $C_{inh}$/mg L$^{-1}$ | $R_s$/Ω cm$^2$ | $Z_{CPE}$ | | $R_p$/Ω cm$^2$ | $C_{dl}$/μF cm$^{-2}$ | $\theta$ | $\eta_E$/% |
| | | | $Y_0$/μΩ$^{-1}$ s$^n$ cm$^{-2}$ | $n$ | | | | |
|---|---|---|---|---|---|---|---|---|
| Blank | 0.0 | 0.12 | 152.5 | 0.724 | 18.7 ± 2 | 336.1 | – | – |
| Ad-Th | 20 | 0.18 | 87.1 | 0.873 | 33.5 ± 3 | 206.9 | 0.441 | 44.1 |
| | 40 | 0.21 | 75.4 | 0.871 | 53.4 ± 5 | 148.8 | 0.649 | 64.9 |
| | 80 | 0.23 | 53.2 | 0.861 | 92.8 ± 8 | 113.3 | 0.798 | 79.8 |
| | 150 | 0.26 | 40.3 | 0.882 | 169.7 ± 12 | 97.1 | 0.889 | 88.9 |
| | 200 | 0.34 | 30.7 | 0.875 | 230.3 ± 18 | 54.0 | 0.918 | 91.8 |
| Fe-Th | 20 | 0.36 | 75.6 | 0.867 | 43.8 ± 4 | 163.2 | 0.573 | 57.3 |
| | 40 | 0.45 | 65.6 | 0.886 | 75.1 ± 6 | 134.8 | 0.751 | 75.1 |
| | 80 | 0.54 | 46.2 | 0.878 | 221.2 ± 15 | 104.1 | 0.915 | 91.5 |
| | 150 | 0.61 | 34.9 | 0.898 | 297.9 ± 23 | 85.6 | 0.937 | 93.7 |
| | 200 | 0.74 | 26.7 | 0.894 | 464.1 ± 34 | 39.6 | 0.959 | 95.9 |

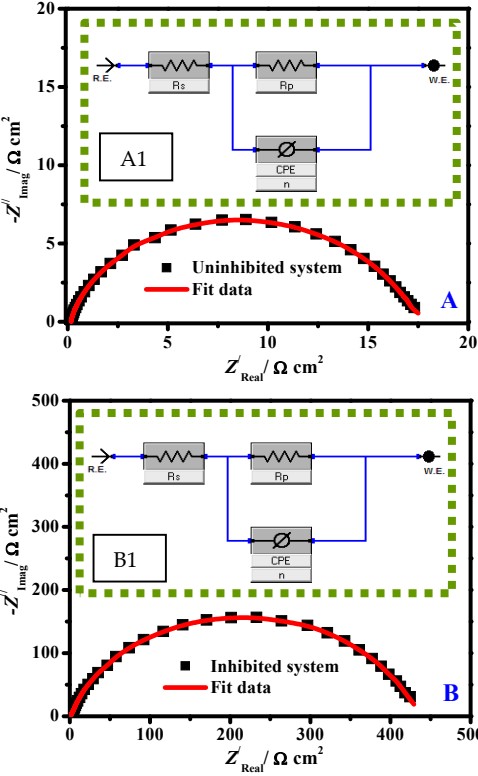

**Figure 5.** Experimental and fitted Nyquist results of C-steel in the uninhibited system (**A**) and inhibited system (**B**). Inset equivalent circuits model for the phenomena (**A1**,**B1**).

### 3.4. Adsorption Considerations

The inhibitor molecule adsorption on the metallic surface could be clarified with the help of an appropriate adsorption isotherm [47]. The adsorption of the corrosion inhibitor molecules Ad-Th and Fe-Th was studied by fitting the experimentally obtained data to several isotherms including Frumkin, Temkin, Langmuir, and Freundlich [48,49]. Amongst the examined isotherms, the Langmuir model exhibited the most suitable fit. The isotherm model can be given as [50]

$$\frac{C_{inh}}{\theta} = C_{inh} + \frac{1}{K_{ads}} \tag{4}$$

Herein, the symbols $C_{inh}$ and $K_{ads}$ signify the inhibitor dose and equilibrium constant for the adsorption, respectively. The curve of $C_{inh}/\theta$ versus $C_{inh}$ generated straight lines as given in Figure 6. The slope and the regression coefficient ($R^2$) are close to unity, which authenticates the successful fitting of the Langmuir model. Based on these explanations it may be conserved that a monolayer of Ad-Th or Fe-Th inhibitors is designed on the metal surface and thus hinders corrosion. The $K_{ads}$ values were calculated from the intercepts of the obtained straight lines (Figure 6) and are as follows: $4.3 \times 10^4$ L mol$^{-1}$ for Ad-Th and $7.8 \times 10^4$ L mol$^{-1}$ for Fe-Th. This consequence could be understood as the Fe-Th adsorption was more efficient due to the increasing adsorption capability at the C-steel/solution interface and hence, the protection capacity was improved. Using the $K_{ads}$ values, the standard free energy of adsorption ($\Delta G^0_{ads}$) was computed using the equation [51]

$$K_{ads} = \frac{1}{55.5} \exp\left(\frac{-\Delta G^0_{ads}}{RT}\right) \tag{5}$$

where the term 55.5 denotes the $H_2O$ concentration in mol/L and the other symbols have their normal meanings. The negative $\Delta G^0_{ads}$ values indicate a strong inhibitors adsorption on the metallic interface

as well as the spontaneity of the adsorption route. Generally, if $\Delta G^0_{ads}$ values are around $-20$ kJ mol$^{-1}$ or lesser is associated with an electrostatic interaction between the charged metal surface and charged inhibitor molecules, i.e., physical adsorption; those of $-40.0$ kJ mol$^{-1}$ or higher involve charge sharing or transfer from the inhibitor molecules to the electrode interface to form a coordinate type bond, i.e., chemical adsorption [44]. The calculated $\Delta G^0_{ads}$ values were $-39.41$ and $-41.03$ kJ mol$^{-1}$ for Ad-Th and Fe-Th, respectively. This suggests that Ad-Th and Fe-Th inhibitors undergo both the chemical and the physical adsorption on the metal surface with a predominance of chemisorption.

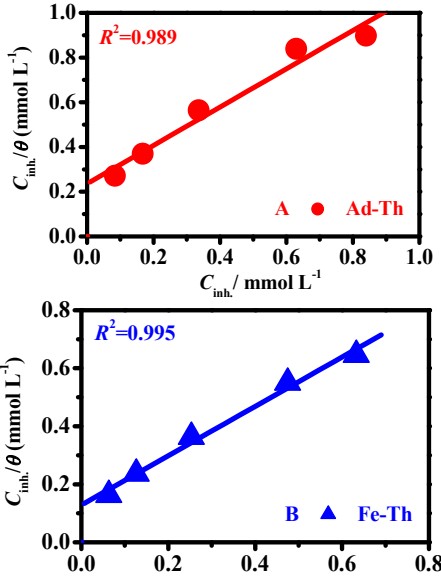

**Figure 6.** Langmuir adsorption diagram for (**A**) Ad-Th and (**B**) Fe-Th on C-steel in molar HCl medium at 50 °C.

### 3.5. Surface Analysis (FESEM and FTIR)

To approve the corrosion protection findings of the as-prepared inhibitors, the surface morphology with and without Ad-Th and Fe-Th inhibitors were detected after 48 h of immersion in the corrosive HCl medium by FE-SEM investigations. Figure 7A,B display the comparison between the surface morphologies of the C-steel specimen before and after 48 h of exposure to 1.0 M hydrochloric acid. It could be displayed the scraping owing to the mechanical preparation of electrode by emery-papers before experiments (Figure 7A). In the uninhibited system, the C-steel surface is severely scratched after immersion with the presence of distortions at the metal surface (Figure 7B). In the existence of an inhibited medium, the C-steel surface is devoid of pit formation or any roughness and very clean (Figure 7C,D). This confirms the development of adsorption film of Ad-Th and Fe-Th molecules covering the C-steel surface, which performances as a barrier towards the overall corrosion process. In the case of the Ad-Th molecule, the C-steel surface was not completely shielded and the protective layer presents some deformities which explain the low protection capacity of this compound. Certainly, the protection capacity of the Ad-Th compound is between 30.9% and 93.6%, and lower than that obtained for the Fe-Th compound (between 38.8% and 97.9%).

The bonding data of crude Ad-Th, as well as the adsorption surface film on the C-steel substrate after exposure to 1.0 M HCl containing 200 ppm of Ad-Th for 48 h, was inspected by FTIR investigation, as exemplified in Figure 8. The spectrum of crude Ad-Th is presented in Figure 8 line A. The bands at 3265 and 3177 cm$^{-1}$ (peaks 1 and 2) are related to the –NH$_2$ and–NH, respectively. The peak at 2909 cm$^{-1}$ (peak 3) is attributed to the C–H aliphatic asymmetric. The band at 1108 cm$^{-1}$ (peak 4) can be assigned to the C=S. By comparing the bands in Figure 8B, it is observed that the common of the peaks of crude Ad-Th are detected in the spectra of the surface layer on C-steel samples immersed in 1.0 M HCl containing 200 ppm of Ad-Th, which approves the existence of Ad-Th in the surface films.

Nonetheless, it is noteworthy that there are some characteristic changes in spectra between the crude Ad-Th and surface film. In the specimen spectrum immersed 1.0 M HCl containing 200 ppm of Ad-Th (Figure 8B), the characteristic bands of –NH$_2$, –NH, and C=S shifted to 3241, 3150, and 1108 cm$^{-1}$, which is different from Figure 8A, which illustrations a probable interaction of the functional groups (–NH$_2$, –NH, and C=S) of Ad-Th inhibitor with the steel surface.

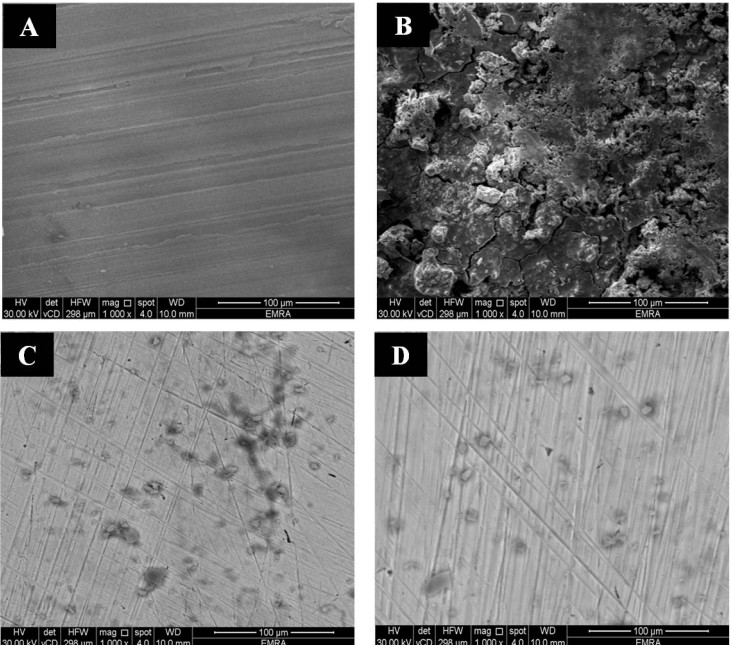

**Figure 7.** SEM picture of C-steel specimens before and after 48 h immersion in 1.0 M HCl at 50 °C. (**A**) before immersion, (**B**) after exposure to blank medium (free inhibitor), (**C**) after immersion with 200 ppm of Ad-Th and (**D**) After immersion with 200 ppm of Fe-Th.

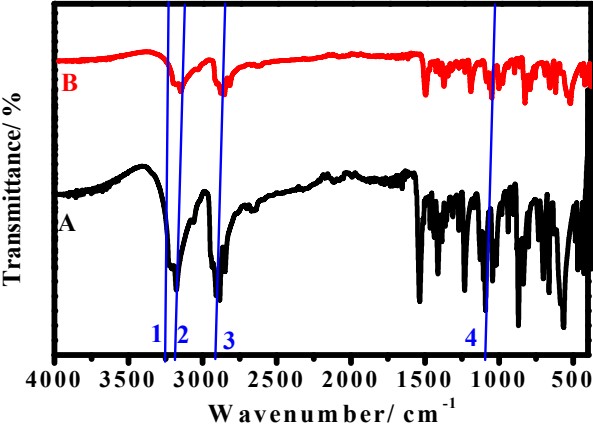

**Figure 8.** FTIR analysis in a range of 400–4000 cm$^{-1}$ of (A) crude Ad-Th and (B) surface film of C-steel sample after immersion in 1.0 M HCl containing 200 ppm of Ad-Th for 48 h at 50 °C.

## 3.6. Corrosion Mitigation Mechanism

The effectiveness of inhibitor additives to diminish corrosion mainly depends on their adsorption capacity. Inhibitor adsorption at the metal/solution interface was affected by various factors such as molecular structure, the existence of donor atoms, the corrosive medium type, nature, and the charge existing on the metal surface. Fe-Th and Ad-Th (thiocarbohydrazones derivatives) are

nitrogen-containing organic molecules (see Scheme 1). In HCl solution, the Fe-Th and Ad-Th inhibitors can occur in protonated and neutral forms as [25]

$$\text{Fe-Th or Ad-Th} + n\text{H}^+ \leftrightarrow [\text{Fe-ThH}_n \text{ or Ad-ThH}_n]^{n+} \tag{6}$$

Furthermore, in HCl solution the potential of zero charge ($E_{\text{PZC}} = 0$) for Fe is −0.59 V [52]. In the current study, $E_{\text{cor}}$ for C-steel in 1.0 HCl solution was −0.523 V. Accordingly, the C-steel surface carries a positive charge because of the $E_{\text{cor}} - E_{\text{PZC}} = +0.067 > 0$ [52]. Consequently, Fe-Th or Ad-Th may be adsorbed on the C-steel surface by the following pathways [52]:

(a) The protonated Fe-Th or Ad-Th molecules might be adsorbed via its electrostatic interactions with the positive charge of the C-steel surface by using Cl⁻ ions attached to the electrode surface as a negative embankment. Then, the donor–acceptor interaction is accomplished between the nitrogen atom of thiocarbohydrazones derivatives and the empty d-orbital of the Fe atoms (or may be connected with the newly produced $Fe^{2+}$ ions at the metal surface) forming Fe-inhibitor complexes as follows:

$$\text{Fe} \rightarrow \text{Fe}^{2+} + 2e^- \tag{7}$$

$$[\text{Fe-ThH}_n \text{ or Ad-ThH}_n]^{n+} + \text{Fe}^{2+} \rightarrow [\text{Fe-ThH}_n\text{-Fe or Ad-ThH}_n\text{-Fe}]^{(2+n)+} \tag{8}$$

(b) The neutral Fe-Th or Ad-Th molecules may be adsorbed at metal/solution interface through the chemisorption route, involving the replacing of $H_2O$ molecules from the electrode surface, where the Fe-Th or Ad-Th additives could be adsorbed at the C-steel interface according to donor–acceptor interactions between π-electrons and/or nitrogen and unoccupied d-orbitals of iron [53] as

$$\text{Fe(H}_2\text{O)}_{\text{ads}} + \text{Fe-Th}_{\text{(aq)}} \text{ or Ad-Th}_{\text{(aq)}} \rightarrow \text{Fe-(Fe-Th)}_{\text{ads}} \text{ or Fe-(Ad-Th)}_{\text{ads}} + \text{H}_2\text{O} \tag{9}$$

Meanwhile, the existence of iron bonded two cyclopentadienyl rings in Fe-Th compared to Ad-Th made the compound more adsorbed to the C-steel surface and so achievement more protection capacity. Moreover, the Fe-Th compound has an electron donor-acceptor conjugated structure and is an effective redox couple of the $Fe^{2+}/Fe^{3+}$ forms [54]. The surface measurements by FE-SEM showed a quantitative improvement in the surface smoothness in the occurrence of the corrosion inhibitor, which supported the adsorption and the inhibition performance. FT-IR studies also supported the presence of the adsorbed inhibitor on the C-steel surface. The above discussion is also consistent with calculated $\Delta G^0_{\text{ads}}$ values, i.e., mixed adsorption type.

*3.7. DFT Calculations*

DFT calculations are an appreciated routine to examine the relationship between the structures of prepared additives and their protection feature. The lowest unoccupied molecular orbital energy ($E_{\text{LUMO}}$) and the highest occupied molecular orbital energy ($E_{\text{HOMO}}$) of the neutral and protonated Ad-Th and Fe-Th molecules using the DMol3 module are presented in Figure 9. The intended quantum chemical parameters of the studied additives are recorded in Table 3. These parameters were found to be very significant to elucidate the chemical reactivity of Ad-Th and Fe-Th additives and its protonated (Ad-ThH and Fe-ThH) form with the metal interface [55]. The energy gap ($\Delta E$) characterizes the reactive inclination of an inhibitor species [55]. Consequently, as the $\Delta E$ becomes lower, the protection capacity would be higher. Fe-Th has a lower $\Delta E$ value. Inhibitor with higher $\Delta E$ values provides lower protection capacity because the energy necessary to eliminate an electron from unavailable orbital will be very high [56]. The trend for $\Delta E$ values have the order Ad-Th > Ad-ThH or Fe-Th > Fe-ThH (Table 3). These findings indicating that the protonated species have a greater tendency to adsorb on the C-steel surface than the non-protonated molecules. The dipole moment ($\mu$) provides an appropriate analysis for the molecule's polarity and also related to the electron distribution in a molecule Inhibitor with a high $\mu$ value interact powerfully with the C-steel interface through dipole–dipole interaction and so measured a respectable corrosion inhibitor. There is no straight relative between $\mu$ of inhibitor

and its protection action [57]. The values of μ are higher for the protonated molecules than for the non-protonated molecules indicating that dipole–dipole interactions are more predominant in the interaction between the C-steel interface and the protonated form than in the interaction between the C-steel surface and the non-protonated form [58].

In this report, the intended μ values are well-coordinated with other quantum parameters—i.e., $E_{HOMO}$, $E_{LUMO}$, $\Delta E$—and the protection capacity increase in the order (Fe-Th > Ad-Th). The number of electrons transferred ($\Delta N$) from compound to C-steel interface is an additional routine to compare quantum indices with the protection capacity [59]. In addition, the values of $\Delta N < 3.6$ suggest that the protection capacity increases as the electron-donating capability of the inhibitor molecules increases [60]. From the findings in Table 3, the values of $\Delta N$ are in the sequence (Ad-ThH < Fe-ThH < Ad-Th < Fe-Th). Herein, Ad-Th and Fe-Th compounds are the donors of electrons, and the C-steel surface is the acceptor. This result supports the emphasis that the adsorption of Ad-Th and Fe-Th additives on the steel interface could take place by donor–acceptor interactions between π-electrons and/or nonbonding electron pairs of the nitrogen atom and unoccupied d-orbitals of iron. According to the above-calculated quantum parameters, it can be established that the protonated Ad-ThH and Fe-ThH forms are more anti-corrosion effective than non-protonated Ad-Th and Fe-Th forms.

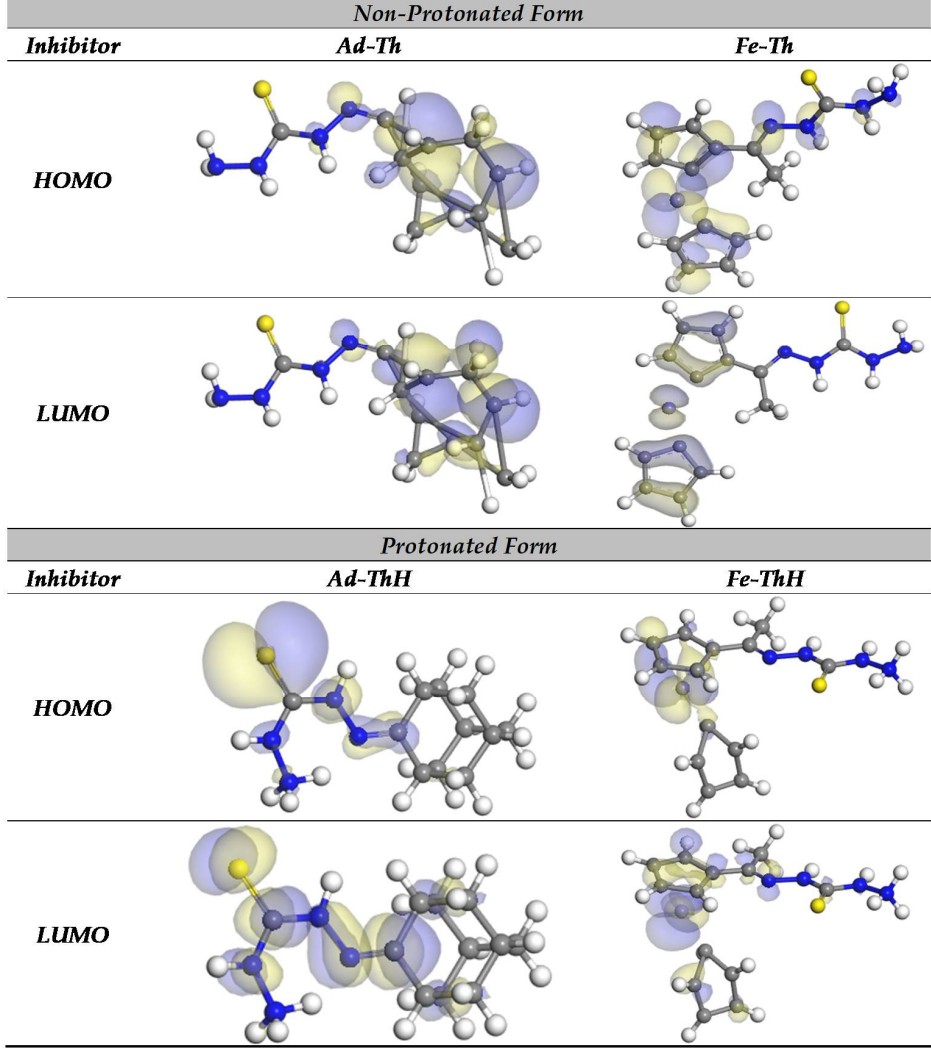

**Figure 9.** HOMO and LUMO of the protonated and non-protonated Ad-Th and Ad-Fe molecules using DMol3 module.

**Table 3.** DFT parameters of the non-protonated and protonated Ad-Th and Fe-Th molecules.

| Parameters | Non-Protonated Form | | Protonated Form | |
|---|---|---|---|---|
| | Ad-Th | Fe-Th | Ad-ThH | Fe-ThH |
| $E_{HOMO}$ (eV) | −3.758 | −3.822 | −3.693 | −3.923 |
| $E_{LUMO}$ (eV) | −2.012 | −3.020 | −2.047 | −3.128 |
| $\Delta E = E_{LUMO} - E_{HOMO}$ (eV) | 1.746 | 0.802 | 1.646 | 0.795 |
| Dipole moments ($\mu$) debye | 7.246 | 5.849 | 13.13 | 22.57 |
| The number of electrons transferred ($\Delta N$) | 1.215 | 1.786 | 0.976 | 1.453 |

*3.8. MC Simulations*

MC simulation has been a favored route to investigate the adsorption shape, and the protective power of corrosion additives adsorbed on a C-steel interface [61]. In order to expect the most promising conformation of the adsorbed inhibitor species on the metal interface and to acquire more understanding into the processes of adsorption, MC simulations were performed for Fe-Th and Ad-Th compounds. The top and side visions of the final adsorption of the Ad-Th and Ad-Fe on the Fe (110) interface in the solution are displayed in Figure 10. The intended binding and interaction energies for both Fe-Th and Ad-Th are recorded in Table 4. As exemplified in Table 4, the interaction energies are negative and of substantial extent, indicating the robust attraction strength between Fe surface and Fe-Th and Ad-Th molecules [62]. Moreover, the higher extent of binding energies designates the greater steadiness of the adsorption route, and convenient adsorption of the as-prepared compounds on the Fe-surface [61,62]. Furthermore, the adsorption energy of Fe-Th is higher than Ad-Th. This indicates that the Fe-Th molecule adsorbs better than the Ad-Th molecule at C-steel/solution interface. Interestingly, these findings are matched with the above DFT and empirical outcomes.

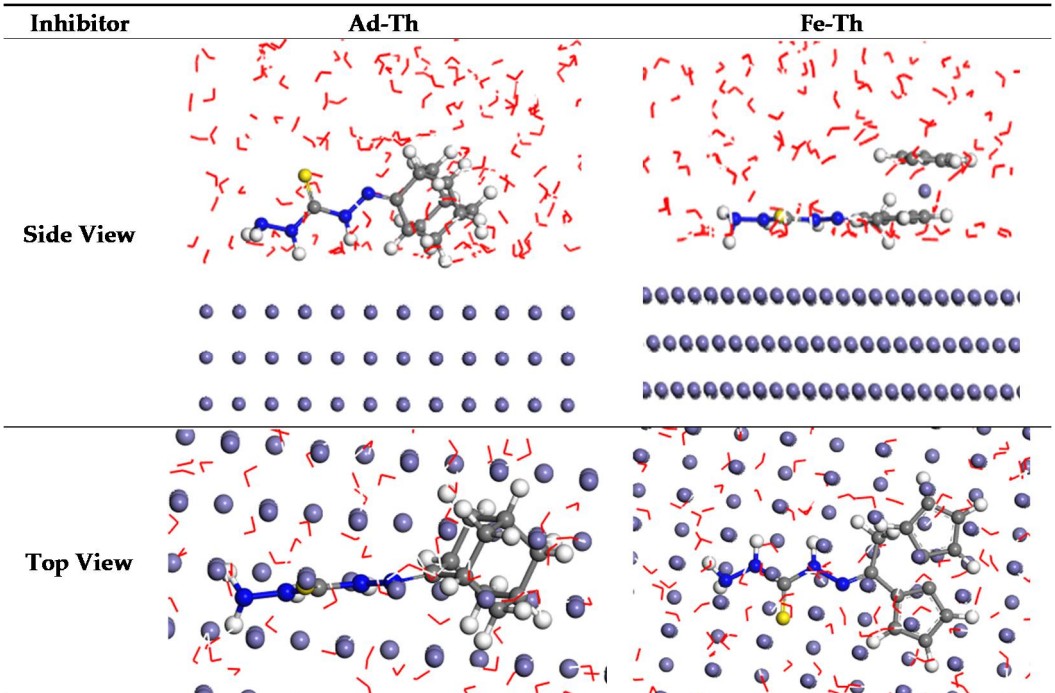

**Figure 10.** Top and side visions of the final adsorption of the Ad-Th and Ad-Fe on the iron (110) surface in solution.

**Table 4.** Selected energy indices acquired from MC simulations for the adsorption of Ad-Th and Fe-Th molecules on Fe (110) surface.

| System | Adsorption Energy/kJ mol$^{-1}$ | Rigid Adsorption Energy/kJ mol$^{-1}$ | Deformation Energy /kJ mol$^{-1}$ | $dE_{ads}/dN_i$: kJ mol$^{-1}$ | $dE_{ads}/dN_i$: Water kJ mol$^{-1}$ |
|---|---|---|---|---|---|
| Fe (110) | | | | | |
| Ad-Th | −2160.84 | −2185.42 | −526.99 | −751.11 | −75.48 |
| H$_2$O | | | | | |
| Fe (110) | | | | | |
| Fe-Th | −3719.59 | −2437.30 | −1282.29 | −1414.63 | −77.87 |
| H$_2$O | | | | | |

*3.9. Comparison of the Protection Capacity of Ad-Th and Fe-Th Inhibitors with Earlier Reports*

The comparison of the protection capacity of the prepared thiocarbohydrazones (Ad-Th and Fe-Th) for the steel corrosion in acidic medium with some selected hydrazone derivatives, previously designated in the same conditions, is given in Table 5 [21–23,63,64]. The list of earlier reports from the naproxen-based hydrazones, dinitrophenyl hydrazone derivatives, propanehydrazide derivatives, benzohydrazide derivatives, and methoxynaphthalen 2-yl propanehydrazide derivatives is registered in Table 5 along with that of the synthesized Ad-Th and Fe-Th molecules [21–23,63,64]. It could be obviously noted that the synthesized Ad-Th and Fe-Th molecules show promising performance compared to the earlier reports with high protection capacity at a meaningfully low inhibitor dose. This supports the practical applicability of the prepared thiocarbohydrazones (Ad-Th and Fe-Th) in the acid pickling process.

**Table 5.** Comparison of the protection capacity of Ad-Th and Fe-Th molecules with the earlier reported findings based on other hydrazone derivatives as inhibitors for steel corrosion in acidic medium.

| Inhibitors | Measurement Method | Inhibitor Concentration | Protection Capacity/% | References |
|---|---|---|---|---|
| Naproxen-based hydrazones | PDP and EIS | 0.1–5.0 mmol/L | 80.2–89.2 | [21] |
| Dinitrophenyl hydrazone derivatives | Weight loss measurements | 0.1–5.0 mmol/L | 84.7–91.2 | [22] |
| Propanehydrazide derivatives | PDP | 0.1–5.0 mmol/L | 63.0–84.1 | [23] |
| Four benzohydrazide derivatives | EIS | 5.0 mmol/L | 84–95 | [63] |
| Methoxynaphthalen 2-yl propanehydrazide derivatives | PDP | 0.1–5.0 mmol/L | 74–91 | [64] |
| Ad-Th | PDP | 20–200 mg/L | 30.9–93.6 | This work |
| Fe-Th | PDP | 20–200 mg/L | 38.8–97.9 | This work |

## 4. Conclusions

The current work reports a thiocarbohydrazones based on adamantane (Ad-Th) and ferrocene (Fe-Th) as novel inhibitors for C-steel corrosion in HCl using thorough experimental and surface morphology analysis supported using computational investigations. The molecules were synthesized by a facile chemical method. The following major conclusion can be drawn out of the present study:

- PDP and EIS measurements revealed an increase in the corrosion inhibition capacity with a rise in the inhibitor dose which reached >97.9% at a dosage of 200 ppm of Fe-Th.
- The inhibitors adsorption on the C-steel followed the isotherm of Langmuir model and the value of $\Delta G^0_{ads}$ indicated the presence of both physical and chemical adsorption modes.
- EIS study designates the occurrence of a one-time constant phenomenon for the inhibitor adsorption in which the polarization resistance increasing with an increment in the inhibitor concentration.

- PDP study exhibited that the adsorption of the inhibitors resulted in a mixed-type with cathodic predominance.
- FESEM measurements showed the appearance of a smoother surface morphology compared to the blank specimen. The FTIR studies supported the adsorption of compound additives on the metallic interface.
- DFT calculations indicate that the protection of C-steel from corrosion is achieved more via the adsorption of protonated Fe-ThH and Ad-ThH forms than that by neutral Fe-Th and Ad-Th forms.
- MC simulation findings displayed that both Fe-Th and Ad-Th molecules adsorb intensely on the iron (110) interface and the inclination of expected binding energies agreed with the empirical protection capacitates.

**Author Contributions:** H.M.A.E.-L. and A.R.S.: conceptualization, supervision, investigation, methodology, resources, formal analysis, data curation, funding acquisition, writing—original draft, writing—review and editing. All authors have read and agreed to the published version of the manuscript.

**Funding:** This research was funded by the Deputyship for Research and Innovation, Ministry of Education in Saudi Arabia.

**Acknowledgments:** The authors extend their appreciation to the Deputyship for Research & Innovation, Ministry of Education in Saudi Arabia for funding this research work through the project number IFT20050.

**Conflicts of Interest:** The authors declare no conflict of interest.

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
