# Peer review of "Thiocarbohydrazones Based on Adamantane and Ferrocene as Efficient Corrosion Inhibitors for Hydrochloric Acid Pickling of C-Steel"

_coatings, doi:10.3390/coatings10111068_

Round 1

Reviewer 1 Report

coatings-984395-peer-review-v1

The paper addresses the corrosion inhibition effect of 2 sythesized organic molecules for carbon steel in hydrochloric acid. It follows a well established standard route in characterizing this effect by electrochemical and surface analytical methods, and aims to support the conclusions by computer simulations.
Basically, the work appears solid and the conclusions appear plausible.
However, the manuscript appears poorly prepared, without care. Consequently, there are a number of flaws, which prevent from following the conclusions and which should not permit publishing the manuscript without revising thoroughly. A detailed list of critical aspects is given below.
Most critical are the references, some of them were identified being without any relation to the text in the manuscript, others do not really provide related information or link further to other papers, which link again. The findings listed below may not be complete, only those were traced down in detail.
The English is acceptable in most parts and the intention of the authors is understandable. However, there appears strange wording, incomplete sentences and mistakes in gramar and spelling. Consequently, this aspect should be edited, too. In particular, "erosion" is frequently confused with "corrosion" (eg lines 282, 380, 403) which is a very irritating mistake in a corrosion related paper. Language editing should be done by a person being familiar with the proper wording in this field of science/technology.

In the following, Lxxx refers to line numbers.

1. L002, 029, 052, 075, 076, 347: Thioohydrazones are a typing mistake but not substances
2. L038: "acid well acidicing" is confusing wording
3. L085: "...the chemicals were used without purification" ..ok, but of which purity were the chemicals? Where did they come from?
4. L168: The procedure of Tafel-evaluation must be explained in detail since there are no obvious "linear segments" in the cuves (Fig. 3). Any particular software used?
5. L174,177,180, 188: These references (34, 35, 36) have NOTHING to do with the context. They are not related at all to the topic (34, 36, 37), or do not provide the basis for the author 's interpretation (35). I did not check all references, but this finding is a strong indication for careless copy-and-paste preparation of this manuscript.
6. L279; Formulas 1,3,4; Tables 1,2: The Greek symbol "Theta" is used at these locations, but this symbol is not explained in any section of the text.
7. Eoc values: Much attention is paid to Eoc. However, there is no comment on the repeatability of these data. This is critical since it is unclear if the variations in the range of just <20 mV (Fig. 2) are experimentally repeatable (i.e. signficant), or just variations in the experiment. Repeatability of the experiments should be commented, in general.
8. L160, 177: This "85 mV criterion" for discriminating anodic/cathodic inhibitors by Eoc is dubious. References 26 and 34 do not provide an explanation for this criterion. 34 is not related to the topic at all. 26 (self cit.) itself also refers to the "85 mV" criterion and refers to its ref. 46; there, this criterion was "introduced" by referencing to its ref. 27; finally there (= Ind. Eng. Chem. Res., 2012, 51, 3966–3974) NO SUCH CRITERION IS MENTIONED. This is a way to create scientific myths! Either provide clear evidence by citing the direct source, or provide a sound explanation, or eliminate this unsupported statement!
9. L257, 258: It is simply WRONG to express the value of the CPE as capacitor value! There is literature on this topic (Corrosion September 2001, Vol. 57, No. 9, pp. 747-748) or use http://www.consultrsr.net/resources/eis/cpe3.htm
10. L287: Ref 51 is WRONG, it is not related to adsorption energy calculations!!
11. L292: it should be -41.03 !! (NEGATIVE value)
12. L292: Explain why these values indicate chemisorption rather than physisorption! This statement is unsupported/unexplained!
Chapter 3.5: Why do we not learn about FTIR of Fe-Th? At least a statement should be provided wether it was also found on the surface or not.
13. L340: Where does the Epcz value for Fe in HCl come from? provide the source!
14. L366: What is "DMol3 module" ?? Reference?
15. L375, 390: These rankings are in CONTRADICTION with the related tables. This creates severe doubts on the explanations and/or data
16. L388: Sentence: "If ..., the protection capacity rise linearly with protection capacity" is pure nonsense.
17. Table 4: The "System" column is corrupted.
18. Table 4: "kcal" is outdated! use kJ as you did in chapter 3.4. Be consistent!
19. Figure 10: What are these little red worm-like items? Explain!

Author Response

Dear Mr. Andrew Yu

Assistant Editor,

We are very excited about the opportunity that has been given to us for revising our manuscript. We have carefully considered all the comments offered by the two reviewers. We would like to extend our sincere appreciation to you and the reviewers for taking the time and effort necessary to provide us such insightful guidance. The revision, based on the review team’s collective input, includes number of positive changes. Based on your guidance, we have accordingly modified the manuscript (IN-TEXT changes are highlighted) and detailed corrections, changes and/or rebuttals against each raised point are listed below with referring to the line and page number for every change/correction (Highlighted in red color).We hope that these revisions improve the paper such that you and the reviewers now deem it worthy of publication in Coatings. Herein, we explain how we revised the paper based on those comments and recommendations and we offer detailed responses to your comments as well as those of the reviewers. Next, we offer detailed responses to the reviewer’s comments:  

RESPONSE TO EDITORS AND REVIEWERS COMMENTS: 

EDITOR COMMENTS:

Please revise the manuscript according to the reviewers' comments and upload the revised file within 8 days. Use the version of your manuscript found at the above link for your revisions, as the editorial office may have made formatting changes to your original submission. Any revisions should be clearly highlighted, for example using the "Track Changes" function in Microsoft Word, so that changes are easily visible to the editors and reviewers. Please provide a cover letter to explain point-by-point the details of the revisions in the manuscript and your responses to the reviewers' comments. Please include in your rebuttal if you found it impossible to address certain comments. The revised version will be inspected by the editors and reviewers. Please detail the revisions that have been made, citing the line number and exact change, so that the editor can check the changes expeditiously. Simple statements like ‘done’ or ‘revised as requested’ will not be accepted unless the change is simply a typographical

error.

Thank you for your great efforts. Your suggestions were taken into account.

REVIEWER # 1 

We would like to thank the reviewer for his great efforts and giving useful criticism to the article. Below are answers to each point.

The paper addresses the corrosion inhibition effect of 2 sythesized organic molecules for carbon steel in hydrochloric acid. It follows a well established standard route in characterizing this effect by electrochemical and surface analytical methods, and aims to support the conclusions by computer simulations. Basically, the work appears solid and the conclusions appear plausible. However, the manuscript appears poorly prepared, without care. Consequently, there are a number of flaws, which prevent from following the conclusions and which should not permit publishing the manuscript without revising thoroughly. A detailed list of critical aspects is given below. Most critical are the references, some of them were identified being without any relation to the text in the manuscript, others do not really provide related information or link further to other papers, which link again. The findings listed below may not be complete, only those were traced down in detail. The English is acceptable in most parts and the intention of the authors is understandable. However, there appears strange wording, incomplete sentences and mistakes in gramar and spelling. Consequently, this aspect should be edited, too. In particular, "erosion" is frequently confused with "corrosion" (eg lines 282, 380, 403) which is a very irritating mistake in a corrosion related paper. Language editing should be done by a person being familiar with the proper wording in this field of science/technology.

Author reply:  We thank the esteemed reviewer for the valuable input.  We have carefully revised the manuscript by Grammarly (https://app.grammarly.com/ddocs/427154829). Moreover, the references were doubly cheeked in the revised manuscript. The word “erosion” was replaced with “corrosion” in the revised manuscript. (See reference list)

In the following, Lxxx refers to line numbers.

  1. L002, 029, 052, 075, 076, 347: Thioohydrazones are a typing mistake but not substances

Thanks for your comment. The word Thiocarbohydrazones was corrected in the revised manuscript.

  1. L038: "acid well acidicing" is confusing wording.

Thanks for your comment. "acid well acidicing"  was corrected to “well acidizing” in the revised manuscript. (See line 38)

  1. L085: "...the chemicals were used without purification" ..ok, but of which purity were the chemicals? Where did they come from?.

Thanks for your comment. All required solvents and chemicals from Aldrich were analytical grade and were used without additional purification.  (See line 83)

  1. L168: The procedure of Tafel-evaluation must be explained in detail since there are no obvious "linear segments" in the cuves (Fig. 3). Any particular software used?

we would like to thank the Reviewer for this comment. Gamry applications include software DC105 for corrosion, EIS300 for EIS measurements, and Echem Analyst 6.0 software package for data fitting. (See line 122).

  1. L174,177,180, 188: These references (34, 35, 36) have NOTHING to do with the context. They are not related at all to the topic (34, 36, 37), or do not provide the basis for the author 's interpretation (35). I did not check all references, but this finding is a strong indication for careless copy-and-paste preparation of this manuscript..

Thanks for your comment. The references (34, 35, 36) were changed by other refernace provides the basis for the interpretation (see Refs. 34-36 in the revised manuscript)

  1. L279; Formulas 1,3,4; Tables 1,2: The Greek symbol "Theta" is used at these locations, but this symbol is not explained in any section of the text..

Thanks for your comment. The Greek symbol "Theta" was explained in the revised manuscript (See lines 186 and 276).

  1. Eoc values: Much attention is paid to Eoc. However, there is no comment on the repeatability of these data. This is critical since it is unclear if the variations in the range of just <20 mV (Fig. 2) are experimentally repeatable (i.e. signficant), or just variations in the experiment. Repeatability of the experiments should be commented, in general.

Thanks for your comment. Each measurement was duplicated at least three times and the values of corrosion parameters recorded as mean ± standard deviation. (See line 128 and Table 1).

  1. L160, 177: This "85 mV criterion" for discriminating anodic/cathodic inhibitors by Eoc is dubious. References 26 and 34 do not provide an explanation for this criterion. 34 is not related to the topic at all. 26 (self cit.) itself also refers to the "85 mV" criterion and refers to its ref. 46; there, this criterion was "introduced" by referencing to its ref. 27; finally there (= Ind. Eng. Chem. Res., 2012, 51, 3966–3974) NO SUCH CRITERION IS MENTIONED. This is a way to create scientific myths! Either provide clear evidence by citing the direct source, or provide a sound explanation, or eliminate this unsupported statement!

We thank the esteemed reviewer for the valuable input. The classification of the organic additives as either a cathodic or an anodic inhibitor includes the change of the OCP values by more than 0.085 V with respect to the OCP value in the blank medium (free inhibitor) [31]. The suggested reference was cited in the revised manuscript for this part (see line 173 and ref. 31)

  1. L257, 258: It is simply WRONG to express the value of the CPE as capacitor value! There is literature on this topic (Corrosion September 2001, Vol. 57, No. 9, pp. 747-748) or use http://www.consultrsr.net/resources/eis/cpe3.htm

We thank the esteemed reviewer for the valuable input.  The values of double layer capacitance were calculated based on EIS300 software and the discussion about this part was revised (See line 273 Table 2).

  1. L287: Ref 51 is WRONG, it is not related to adsorption energy calculations!!

Thanks for your comment. The reference 51 was changed in the revised manuscript (See Ref. 51).

  1. L292: it should be -41.03 !! (NEGATIVE value)

Thanks for your comment. Noted and corrected in the revised manuscript (see line 310)

  1. L292: Explain why these values indicate chemisorption rather than physisorption! This statement is unsupported/unexplained!

Thanks for your comment. Generally, if values are around -20 kJ mol−1 or lesser is associated with an electrostatic interaction between charged metal surface and charged inhibitor molecules, i.e. physical adsorption; those of -40.0 kJ mol−1 or higher involve charge sharing or transfer from the inhibitor molecules to the electrode interface to form a coordinate type bond i.e. chemical adsorption [44]. The calculated values were -39.41 kJ mol-1 and -41.03 kJ mol-1 for Ad-Th and Fe-Th, respectively. This suggests that Ad-Th and Fe-Th inhibitors undergo both the chemical and the physical adsorption on the metal surface with a predominance of chemisorption.  (See line 306 and ref. 44)

  1. Chapter 3.5: Why do we not learn about FTIR of Fe-Th? At least a statement should be provided wether it was also found on the surface or not.

Thanks for your comment. The surface morphology by FTIR was studied in the presence of Ad-Th compound as a model. Similar behaviour was obtained in the presence of Fe-Th.

  1. L340: Where does the Epcz value for Fe in HCl come from? Provide the source!.

Thanks for your comment. The Epcz value for Fe in HCl was obtained from the previous study. The new reference was cited in the revised manuscript (see Ref. 52)

  1. L366: What is "DMol3 module" ?? Reference?

Thanks for your comment. More details were added in the revised manuscript. DFT calculations and MC simulation had been conducted using DMol3 and adsorption Locator modules in Materials Studio software V.7.0 from Accelrys Inc. USA. For DFT calculations, the investigated Ad-Th and Fe-Th molecules have been fully optimized using B3LYP functional (Becke-3-Parameters-Lee-Yang-Parr) with the DNP basis set and permitting the treating of solvation impacts utilizing COSMO controls, all these inputs are well-defined as in Eid et al, 2020 [28]. For MC simulation, the adsorption locator reveals the potential adsorption configurations of the Ad-Th and Fe-Th molecules with Monte Carlo searches on the Fe (1 1 0) surface to assess the inhibition capacity of additives [29]. The interactions of Ad-Th and Fe-Th and Fe surface (1 1 0) accomplished in a simulation box (32.27Å×32.27Å×50.18Å) with periodic boundary conditions [29]. Forcite classical simulation engine was utilized to optimize the energy of Ad-Th and Fe-Th molecules. To the establishment, the Ad-Th and Fe-Th molecules/C-steel corrosion system in aqueous media the layer builder was implemented, and this system includes the optimized Fe (1 1 0) surface, water, and the inhibitor molecules. The universal simulation studies with force field were operated to simulate the adsorption performance of Ad-Th and Fe-Th molecules on the surface of iron (1 1 0) [30]. (see line 136 and Refs. 28-30)

  1. L375, 390: These rankings are in CONTRADICTION with the related tables. This creates severe doubts on the explanations and/or data.

We thank the esteemed reviewer for the valuable input and we apologize for this confusion. Noted and revised. (See lines 390-403)

  1. L388: Sentence: "If ..., the protection capacity rises linearly with protection capacity" is pure nonsense.

We thank the esteemed reviewer for the valuable input. The values of ΔN < 3.6 suggest that the protection capacity increases as the electron donating capability of the inhibitor molecules increases [60]. (See line 407)

  1. Table 4: The "System" column is corrupted.

We thank the esteemed reviewer for the valuable input. Noted and corrected. (See Table 4)

  1. Table 4: "kcal" is outdated! use kJ as you did in chapter 3.4. Be consistent!.

Thanks for your comment. The unit kJ mol−1 was used in the revised manuscript. (See Table 4)

  1. Figure 10: What are these little red worm-like items? Explain!

Thanks for your comment. From Table 4 the corrosion system was Fe/inhibitor/water. The red worm is water molecule.

Again, we appreciate the opportunity to revise our work for consideration for publication in Corrosion Science. We hope our revision meet your approval. We believe that each of the reviewers did an excellent job of commenting on the manuscript, and we want to extend our appreciation for taking the time and effort necessary to provide such insightful guidance. We are looking forward to your positive response.   

Sincerely,  

Hany Mohamed Abd El-lateef

Reviewer 2 Report

Manuscript ID nanomaterials JTICE-D-20-01316 entitled:

Thiocrabohydrazones based on adamantane and 3 ferrocene as efficient corrosion inhibitors for 4 hydrochloric acid pickling of C-steel

General comment

The work examined the corrosion protection performance of two thiocrabohydrazones based on adamantane and ferrocene in hydrochloric acid medium for C76 steel. PDP and EIS FESEM, FTIR techniques and density functional theory (DFT) and Monte Carlo (MC) simulations was employed to validate their experimental results.

  1. Other research focused on thiocrabohydrazone as corrosion inhibitors that appeared in literature can be mentioned in Introduction part.
  2. In Experimental part. Please give complete explanation upon computational methods. Seeking into the mentioned references [28-30], no information is given for MC.
  3. What is the number of replicates for each test? 
  4. Because the experiments are performed at 50 °C, mentioned this condition in all figures, tables and conclusions. (i.e. Figure 2. Variation of the OCP of C- steel in molar HCl, with immersion time in the lack and occurrence of 200 ppm of Ad-Th and Fe-Th inhibitors at 50°C).
  5. It is not mandatory, but some information about protection against corrosion at ambient temperature is worth mentioning, at least the θ values for the studied compounds, for comparison.
  6. Please check if Table 4 is complete. On page 18 (R407) the intended binding and interaction energies for both Fe-Th and Ad-Th are mention, but appear only for Ad-Th. Correct or complete the table in accordance with the text.
  7. Comparison of obtained results with other best reported for other inhibitors in the same condition, particularly at 50 °C is needed. Provide a table comparing the performance of you compounds with the other reported in literature.

Author Response

Dear Mr. Andrew Yu

Assistant Editor,

We are very excited about the opportunity that has been given to us for revising our manuscript. We have carefully considered all the comments offered by the two reviewers. We would like to extend our sincere appreciation to you and the reviewers for taking the time and effort necessary to provide us such insightful guidance. The revision, based on the review team’s collective input, includes number of positive changes. Based on your guidance, we have accordingly modified the manuscript (IN-TEXT changes are highlighted) and detailed corrections, changes and/or rebuttals against each raised point are listed below with referring to the line and page number for every change/correction (Highlighted in red color).We hope that these revisions improve the paper such that you and the reviewers now deem it worthy of publication in Coatings. Herein, we explain how we revised the paper based on those comments and recommendations and we offer detailed responses to your comments as well as those of the reviewers. Next, we offer detailed responses to the reviewer’s comments:  

RESPONSE TO EDITORS AND REVIEWERS COMMENTS: 

EDITOR COMMENTS:

Please revise the manuscript according to the reviewers' comments and upload the revised file within 8 days. Use the version of your manuscript found at the above link for your revisions, as the editorial office may have made formatting changes to your original submission. Any revisions should be clearly highlighted, for example using the "Track Changes" function in Microsoft Word, so that changes are easily visible to the editors and reviewers. Please provide a cover letter to explain point-by-point the details of the revisions in the manuscript and your responses to the reviewers' comments. Please include in your rebuttal if you found it impossible to address certain comments. The revised version will be inspected by the editors and reviewers. Please detail the revisions that have been made, citing the line number and exact change, so that the editor can check the changes expeditiously. Simple statements like ‘done’ or ‘revised as requested’ will not be accepted unless the change is simply a typographical

error.

Thank you for your great efforts. Your suggestions were taken into account.

REVIEWER # 2 

We would like to thank the reviewer for his criticism to the article. Below are answers to each point.

The work examined the corrosion protection performance of two thiocrabohydrazones based on adamantane and ferrocene in hydrochloric acid medium for C76 steel. PDP and EIS FESEM, FTIR techniques and density functional theory (DFT) and Monte Carlo (MC) simulations were employed to validate their experimental results.

  1. Other research focused on thiocrabohydrazone as corrosion inhibitors that appeared in literature can be mentioned in Introduction part.

Thanks for your fruitful comment.  The reaction of malononitrile dimer with thiocarbohydrazones gave thiadiazines derivatives based on microwave-assisted as green chemistry [18]. Syntheses of coumarin by using simple and efficient methods are prepared as fluorescent chemosensor for fluoride detection [19]. Reaction of bisthiocarbohydrazones with cis-dioxomolybdenum(VI) gave complex for electrochemical application [20]. Enhanced corrosion inhibition of carbon steel in HCl solution by synthesized hydrazone derivatives was reported [21-24]. (See lines 65-70 and Refs. 21-24)

  1. In Experimental part. Please give complete explanation upon computational methods. Seeking into the mentioned references [28-30], no information is given for MC.

we would like to thank the Reviewer for this comment. More details were added in the revised manuscript. DFT calculations and MC simulation had been conducted using DMol3 and adsorption Locator modules in Materials Studio software V.7.0 from Accelrys Inc. USA. For DFT calculations, the investigated Ad-Th and Fe-Th molecules have been fully optimized using B3LYP functional (Becke-3-Parameters-Lee-Yang-Parr) with the DNP basis set and permitting the treating of solvation impacts utilizing COSMO controls, all these inputs are well-defined as in Eid et al, 2020 [28]. For MC simulation, the adsorption locator reveals the potential adsorption configurations of the Ad-Th and Fe-Th molecules with Monte Carlo searches on the Fe (1 1 0) surface to assess the inhibition capacity of additives [29]. The interactions of Ad-Th and Fe-Th and Fe surface (1 1 0) accomplished in a simulation box (32.27Å×32.27Å×50.18Å) with periodic boundary conditions [29]. Forcite classical simulation engine was utilized to optimize the energy of Ad-Th and Fe-Th molecules. To the establishment, the Ad-Th and Fe-Th molecules/C-steel corrosion system in aqueous media the layer builder was implemented, and this system includes the optimized Fe (1 1 0) surface, water, and the inhibitor molecules. The universal simulation studies with force field were operated to simulate the adsorption performance of Ad-Th and Fe-Th molecules on the surface of iron (1 1 0) [30]. (see line 136 and Refs. 28-30)

  1. What is the number of replicates for each test?

Thanks for your comment. Each measurement was duplicated at least three times and the values of corrosion parameters recorded as mean ± standard deviation. (See line 128).  

  1. Because the experiments are performed at 50 °C, mentioned this condition in all figures, tables and conclusions. (i.e. Figure 2. Variation of the OCP of C- steel in molar HCl, with immersion time in the lack and occurrence of 200 ppm of Ad-Th and Fe-Th inhibitors at 50 °C).

Thanks for your comment. Noted and done in the revised manuscript. (See Tables and Figures captions)

  1. It is not mandatory, but some information about protection against corrosion at ambient temperature is worth mentioning, at least the θ values for the studied compounds, for comparison.

Thanks for your fruitful comment. It is well documented that the corrosion rate of C-steel depends on the temperature, and the maximum corrosion rate is noticed at 50 °C, the optimized corrosion temperature [26]. Consequently, to evaluate the protection capacity of the as-prepared Fe-Th and Ad-Th additives on C-steel, all the measurements were accomplished at 50 °C. (See line 125). The prepared compounds showed excellent protection capacity at 50 °C.

  1. Please check if Table 4 is complete. On page 18 (R407) the intended binding and interaction energies for both Fe-Th and Ad-Th are mention, but appear only for Ad-Th. Correct or complete the table in accordance with the text.

Thanks for your fruitful comment. Noted and corrected. (See Table 4)

  1. Comparison of obtained results with other best reported for other inhibitors in the same condition, particularly at 50 °C are needed. Provide a table comparing the performance of you compounds with the other reported in literature.

Thanks for your fruitful comment. The comparison of the protection capacity of Ad-Th and Fe-Th inhibitors with earlier reports was studied in the revised manuscript. (See section 3.9 line 444 and Table 5)

Again, we appreciate the opportunity to revise our work for consideration for publication in Corrosion Science. We hope our revision meet your approval. We believe that each of the reviewers did an excellent job of commenting on the manuscript, and we want to extend our appreciation for taking the time and effort necessary to provide such insightful guidance. We are looking forward to your positive response.   

Sincerely,  

Hany Mohamed Abd El-lateef

Round 2

Reviewer 1 Report

The efforts of the authors in revising the manuscript have definitely improved the quality of the paper and it appears almost ready to publish.

However, there is one item, which is still unsatisfactory to me, as interested reader, and as reviewer. This item is:
-------------------------------------------
8. L160, 177: This "85 mV criterion" for discriminating anodic/cathodic inhibitors by Eoc is dubious. References 26 and 34 do not provide an explanation for this criterion. 34 is not related to the topic at all. 26 (self cit.) itself also refers to the "85 mV" criterion and refers to its ref. 46; there, this criterion was "introduced" by referencing to its ref. 27; finally there (= Ind. Eng. Chem. Res., 2012, 51, 3966–3974) NO SUCH CRITERION IS MENTIONED. This is a way to create scientific myths! Either provide clear evidence by citing the direct source, or provide a sound explanation, or eliminate this unsupported statement!

Author reply: We thank the esteemed reviewer for the valuable input. The classification of the organic additives as either a cathodic or an anodic inhibitor includes the change of the OCP values by more than 0.085 V with respect to the OCP value in the blank medium (free inhibitor) [31]. The suggested reference was cited in the revised manuscript for this part (see line 173 and ref. 31)
------------------------------------------
Remark: The reference was not suggested, in fact it was classified an unsuitable source.

I have visited [31] again and I have again NOT FOUND any indication/explanation for this "0.085 mV" criterion in this paper (= Ind. Eng. Chem. Res., 2012, 51, 3966–3974)
From my point of view, I have no idea how such a criterion could be justified by scientific or technical arguments. So, I would like to know exactly the justification.
In fact, I have the impression that this criterion was established by mis-interpretations and transfers from one paper to another by citing (see my previous comment above).

As reviewer, I consider it my duty to insist in this question since we should avoid creating and keeping alive unjustified "general laws".
Therefore, I may suggest again:
Either provide clear evidence by citing the direct source of explanation for this particular criterion, or provide a sound explanation yourself (why exactly 85 mV??), or eliminate this statement since unsupported (which would not at all impair the quality of your paper).

Author Response

Dear Mr. Andrew Yu

Assistant Editor,

Firstly we would like to thank you for giving us a chance to resubmit the paper, and also thank the reviewers for giving us constructive suggestions which would help us to improve the quality of the paper. Here we submit a new version of our manuscript, which has been modified according to the reviewers’ suggestions. We mark the changes in red color in the new version manuscript. The detailed corrections are listed below point by point: 

RESPONSE TO EDITORS AND REVIEWERS COMMENTS: 

EDITOR COMMENTS:

Please revise the manuscript according to the reviewers' comments and upload the revised file within 2 days. Use the version of your manuscript found at the above link for your revisions, as the editorial office may have made formatting changes to your original submission. Any revisions should be clearly highlighted, for example using the "Track Changes" function in Microsoft Word, so that changes are easily visible to the editors and reviewers. Please provide a cover letter to explain point-by-point the details of the revisions in the manuscript and your responses to the reviewers' comments. Please include in your rebuttal if you found it impossible to address certain comments. The revised version will be inspected by the editors and reviewers. Please detail the revisions that have been made, citing the line number and exact change, so that the editor can check the changes expeditiously. Simple statements like ‘done’ or ‘revised as requested’ will not be accepted unless the change is simply a typographical

error.

Thank you for your great efforts. Your suggestions were taken into account.

REVIEWER # 1 

We would like to thank the reviewer for his great efforts and giving useful criticism to the article. Below are answers to each point.

The efforts of the authors in revising the manuscript have definitely improved the quality of the paper and it appears almost ready to publish.

However, there is one item, which is still unsatisfactory to me, as interested reader, and as reviewer. This item is:
-------------------------------------------
8. L160, 177: This "85 mV criterion" for discriminating anodic/cathodic inhibitors by Eoc is dubious. References 26 and 34 do not provide an explanation for this criterion. 34 is not related to the topic at all. 26 (self cit.) itself also refers to the "85 mV" criterion and refers to its ref. 46; there, this criterion was "introduced" by referencing to its ref. 27; finally there (= Ind. Eng. Chem. Res., 2012, 51, 3966–3974) NO SUCH CRITERION IS MENTIONED. This is a way to create scientific myths! Either provide clear evidence by citing the direct source, or provide a sound explanation, or eliminate this unsupported statement!

We thank the esteemed reviewer for the valuable input. The classification of the organic additives as either a cathodic or an anodic inhibitor includes the change of the OCP values by more than 0.085 V with respect to the OCP value in the blank medium (free inhibitor) [31]. The suggested reference was cited in the revised manuscript for this part (see line 173 and ref. 31)
------------------------------------------
Remark: The reference was not suggested, in fact it was classified an unsuitable source.

I have visited [31] again and I have again NOT FOUND any indication/explanation for this "0.085 mV" criterion in this paper (= Ind. Eng. Chem. Res., 2012, 51, 3966–3974)
From my point of view, I have no idea how such a criterion could be justified by scientific or technical arguments. So, I would like to know exactly the justification.
In fact, I have the impression that this criterion was established by mis-interpretations and transfers from one paper to another by citing (see my previous comment above).

As reviewer, I consider it my duty to insist in this question since we should avoid creating and keeping alive unjustified "general laws".
Therefore, I may suggest again:
Either provide clear evidence by citing the direct source of explanation for this particular criterion, or provide a sound explanation yourself (why exactly 85 mV??), or eliminate this statement since unsupported (which would not at all impair the quality of your paper).

We thank the esteemed reviewer for the valuable input.  The statement was deleted in the revised manuscript (See Line 170)

Again, we appreciate the opportunity to revise our work for consideration for publication in Corrosion Science. We hope our revision meet your approval. We believe that each of the reviewers did an excellent job of commenting on the manuscript, and we want to extend our appreciation for taking the time and effort necessary to provide such insightful guidance. We are looking forward to your positive response.   

Sincerely,  

Hany Mohamed Abd El-lateef